# Poly(carbazole-co-1,4-dimethoxybenzene): Synthesis, Electrochemiluminescence Performance, and Application in Detection of Fe^3+^

**DOI:** 10.3390/polym14153045

**Published:** 2022-07-27

**Authors:** Pengchong Hou, Xian Zhang, Qian Lu, Shunwei Chen, Qiang Liu, Congde Qiao, Hui Zhao

**Affiliations:** 1School of Materials Science and Engineering, Qilu University of Technology, Shandong Academy of Sciences, Jinan 250353, China; hou17806022585@163.com (P.H.); swchen@qlu.edu.cn (S.C.); yeah21138@126.com (Q.L.); cdqiao@qlu.edu.cn (C.Q.); 2School of Chemical Engineering, Sichuan University, Chengdu 610065, China

**Keywords:** electrochemiluminescence, polycarbazole, ECL sensor, detection, Fe^3+^

## Abstract

In this study, four polycarbazole derivatives (PCMB-Ds) with different alkyl side chains were designed and synthesized via Wittig–Horner reaction. A novel solid-phase electrochemiluminescence (ECL) system was prepared by immobilizing PCMB-D on an indium tin oxide (ITO) electrode with polyvinylidene fluoride (PVDF) in the presence of tripropylamine (TPrA). It could be found that the increase in alkyl side chain length had little effect on the ECL signal of PCMB-D, while the increase in the degree of polymerization (DP) greatly enhanced the ECL signal. Furthermore, the P-3/ITO ECL sensor based on the polyoctylcarbazole derivative (P-3) with the best ECL performance was successfully constructed and detected Fe^3+^ under the optimal experimental conditions. The ECL signal steadily diminished with the increased concentration of Fe^3+^ because of the competition and complexation between Fe^3+^ and P-3 under the condition of pH 7.4. This P-3/ITO platform could realize a highly sensitive and selective detection of Fe^3+^ with a wide detection range (from 6 × 10^−8^ mol/L to 1 × 10^−5^ mol/L) and low detection limit of 2 × 10^−8^ mol/L, which could allow the detection of Fe^3+^ in multiple scenarios, and would have a great application prospect.

## 1. Introduction

Fe^3+^ is essential for the balance of various environmental and biological systems [1]. In the human body, Fe^3+^ has unique significance in oxygen transport and metabolism [2], cellular enzymatic reactions [3], DNA repair, etc. [4]. The content of Fe^3+^ should be less than 0.3 mg/L in the environment, 5.36 μmol/L in drinking water, and 70~150 μg/L in serum [5]. Deficiency or excessive use of Fe^3+^ can disrupt normal biological processes, leading to anemia, decreased immunity, infection, Parkinson’s, Alzheimer’s, cancer, etc. Consequently, the rapid detection and real-time monitoring of the concentration of Fe^3+^ are of great significance to protect the ecological environment and people’s lives and health. Therefore, the development of suitable detecting methods and sensors for the detection and monitoring of the concentration of Fe^3+^ has received extensive attention.

At present, the main methods for detecting Fe^3+^ are atomic absorption spectroscopy [6], gas chromatography, and ultraviolet–visible spectrophotometry [7], but their detection process is complicated [8] and the equipment is expensive and requires relatively skilled professionals [9], while the ECL analysis in the electrochemical sensor method has developed rapidly because of its convenient and fast operation, high sensitivity [10] and low cost [11], and this method has attracted more and more attention [12]. Therefore, it is very necessary and promising to design new ECL reagents to detect Fe^3+^.

Although the ECL method has already been used, it has only been used in the analysis and testing field in recent years [13]. ECL analysis [14] is the latest product of the combination of chemiluminescence, electrochemistry, bioanalysis, microelectronic technology [15], and sensing technology, showing the unique advantages of high sensitivity, simple operation, strong controllability, quick and easy analysis [16] and great potential in the field of clinical [17], agricultural [18], and environmental monitoring [19]. In terms of ECL active materials, Ru(bpy)_3_^2+^ [20], Luminol [21], iridium complexes [22], quantum dots [17], and other inorganic materials [23] have relatively stable luminescence and mature research, but their luminescence brightness and lifetime need to be improved, and their excitation potential is high [9]. Recently [24], CPs have attracted the attention of many researchers. CPs had always been regarded as insulating materials until Burroughes of Cambridge University first observed the ECL signal of poly (phenylene vinylene) (PPV) in 1990 [25], which opened a new era in the research of CPs as ECL materials. Since then, more and more polymer ECL active materials such as polypyrrole (PPY) [26], polyvinyl carbazole (PVK) [27], polythiophene (PT) [28] and polyfluorene (PF) were gradually observed and exhibited unique optical and electrical properties, such as good expandability, strong light trapping ability, low energy consumption, high fluorescence quantum yield, great biocompatibility, low price, and other unmatched advantages [29]. However, most of the traditional CPs used for ECL bioassays were liquid phase and might bind non-specifically with other substances in the aqueous phase, which would affect the detection accuracy. Compared with the liquid phase, the luminescent reagent of the solid-phase system was separated from the object to be detected, improving detection accuracy [30] and saving the consumption of CPs. Therefore, it is very meaningful to design a solid-phase ECL sensing platform based on a novel CP ECL material to realize the detection of Fe^3+^.

Polycarbazole [31] is a linear conjugated polymer with an aromatic heterocyclic ring containing nitrogen atoms. Its conjugated structure is extended by a benzene ring, which is similar to a molecular wire, giving electrical conductivity [32]. The carbazole ring located in the main part of the polymer is conducive to carrier migration. In addition, the three-dimensional structure of polycarbazole provides excellent solubility and film-forming properties, high concentration of active groups, good signal amplification performance, and chemical stability [33]. More importantly, their molecular structure and surface functional groups are easily designed and tailored. For example, carbazole can easily be introduced to a variety of groups on the nitrogen atom, and modified at the position of third and sixth, so it is a very promising ECL material. In addition, the excited state generated by the electrification of polycarbazole may interact with strong oxidation of Fe^3+^, which is likely to affect the ECL signal of the system, thereby this system may be applied to detect Fe^3+^.

In addition, the ECL signals are closely related to the structure of CPs. The proper CP structure will help to improve the sensitivity of ECL biosensors. Herein, poly(ethylcarbazole-co-1,4-dimethoxybenzene) (P-1), poly(butylcarbazole-co-1,4-dimethoxybenzene) (P-2), poly(octylcarbazole-co-1,4-dimethoxybenzene) (P-3) and poly(hexadecylcarbazole-co-1,4-dimethoxybenzene) (P-4) were successfully synthesized based on Wittig–Horner reaction. This paper also discusses the relationship between the ECL signal and the structure of PCMB-D with different alkyl side chains and different DPs. In addition, PVDF acted as a binder between PCMB-D and ITO to obtain a more stable ECL signal. Furthermore, a solid-phase ECL system of PCMB-D on the electrode for detecting Fe^3+^ was constructed, and the detection mechanism was investigated. In our case, PCMB-D is expected to be a type of novel ECL reagent to impel the development of the CP ECL analysis method.

## 2. Materials and Methods

### 2.1. Materials

The ITO used in this experiment was purchased from Liaoning Huite Optoelectronics Technology Co, Ltd., Anshan City, China with a thickness of 0.7 mm, a transmittance of 85%, a resistance of 6–7 ohms, and a temperature resistance of 320 °C. Ethyl acetate (EtOAc), tetrahydrofuran (THF), dimethylketone (DMK), methyl alcohol (MT), *N, N*-dimethylformamide (DMF), acetonitrile (AN), *N*-methylpyrrolidone (NMP), PVDF and other chemicals were bought from commercial suppliers. All solvents used in this paper were all analytical purity reagents. Ultrapure water was used throughout the experiments. It is worth mentioning that the electrode used in this experiment is ITO conductive glass. The ECL signal in this experiment is positively correlated with the aggregation state and coverage area of the polymer, and the larger modifiable area of ITO is convenient for the collection of ECL signals. In addition, the transparent electrode facilitates observations of the film formation status on the site.

### 2.2. Synthesis of PCMB-D

The synthetic procedure of a series of PCMB-Ds containing the same conjugated backbone with different alkyl side chains is shown in Figure 1a. 9H-carbazole-3,6-dicarbaldehyde (M1), 3,6-diformyl-9-butylcarbazole (M2), 3,6-diformyl-9-octylcarbazole (M3), and 3,6-diformyl-9-hexadecylcarbazole (M4) are synthesized according to the previous literature [34]. P-1, P-2, P-3, and P-4 are synthesized via Wittig–Horner by combining 2,5-di-(ethoxy phosphoryl methylene)-1,4-dimethoxybenzene (M5) and M1, M2, M3, and M4, respectively. The relevant ^1^H NMR spectra, ^13^C NMR spectra, and FTIR are observed in Appendix A. Gel permeation chromatographic (GPC) data are shown in Table 1.

### 2.3. Measurement

^1^H NMR (400 MHz) spectra and ^13^C NMR (101 MHz) spectra were recorded on a Bruker Avance 400 spectrometer in deuterated chloroform (CDCl_3_) and dimethyl sulfoxide (DMSO) with tetramethylsilane (TMS) internal standard as reference. FTIR data for the PCMB-D were obtained using a Nicolet iS10 spectrometer (Thermo Fisher Scientific Inc., Waltham, MA, USA) equipped with an attenuated total reflection (ATR) accessory. The FTIR spectra were recorded in the range of 400 to 4000 cm^−1^ with a resolution of 4 cm^−1^ and 16 scans. The data were analyzed with Omnic software. The molecular weights and polymer dispersity indexes (PDI = Mw/Mn) were measured with PL-120 model permeation gel chromatography in DMF at room temperature using a calibration curve of polystyrene standard. The melting point (Mp) of the monomers was obtained by the RD-1 melting point tester. Thermal gravimetric analysis (TGA, Mettler Toledo, Switzerland) was used to test the thermal stability of the PCMB-D. TGA was carried out in N_2_ atmosphere from 100 °C to 800 °C with a heating rate of 10 °C/min. Ultraviolet–visible (UV–Vis) absorption and fluorescence (FL) spectra were collected on a Shimadzu UV-2500 spectrophotometer and Hitachi F-7000 Fluorescence Photometer, respectively. Cyclic voltammetry (CV) curves and ECL signals were recorded on the MPI-E analysis system of Xi’an Remax Analytical Instrument Co, Ltd., Xi’an, China, equipped with a platinum wire and an Ag/Agcl as the auxiliary electrode and reference electrode, respectively.

### 2.4. Fabrication of ECL Sensor

Prior to fabrication of the ECL system, the ITO glass electrodes with a length of 2 cm and width of 1 cm were immersed in acetone, ethanol, and ultrapure water, and ultrasonically cleaned for 15 min. Then, the cleaned ITO was dried under a stream of N_2_ for usage. PCMB-D and PVDF were dissolved in NMP together. The mixture was sonicated for 30 min, and then ball milled at 434 rpm for 4 h at room temperature. The ball-milled sample was coated immediately and uniformly on the dry ITO, and then baked at 90 °C for 12 h in a vacuum state. The modified ITO as a working electrode was immersed into 0.02 mol/L TPrA (0.1 mol/L PBS, pH = 7.4) for Fe^3+^ detection. The high voltage and scan rates were 800 V and 1 V/s, respectively.

## 3. Results and Discussion

### 3.1. Synthesis and Characterization

PCMB-D were synthesized via Wittig–Horner [34] as described in Figure 1a. The structures of the polymers were characterized by ^1^H NMR, ^13^C NMR, and FTIR (Appendix A). In the case of P-3, as depicted in Appendix A, the chemical shift at δ = 10.12 ppm was assigned to the aldehyde group of M3, and it completely disappeared in the ^1^H NMR spectrum of P-3. In addition, the chemical shift at δ = 6.78 ppm in ^1^H NMR of Appendix A, the absorption spectrum at 123 ppm in the ^13^C NMR of Appendix A, and the stretching vibration and flexural vibration of C=C at 1600 cm^−1^ and 798 cm^−1^ in FTIR are characteristic of C=C forming. In addition, the stretching vibration of C-O at 1208 cm^−1^, alkyl stretching vibrations of C-H at 2929 cm^−1^, and the stretching vibrations of C-H of aromatic rings (1601 cm^−1^) further proved that P-3 was successfully prepared by Wittig–Horner reaction based on M3 and M5. The similar deductions of other polymers could be obtained by comparing the ^1^H NMR, ^13^C NMR, and FTIR spectra of other monomers with their target polymers.

Based on the large delocalization of polycarbazole, a charge transfer bridge of C=C bond and a strong electron-donating group -OCH_3_ were introduced, which provide delocalized transfer conditions for electrons and make the PCMB-D exhibit unique optical and electrical properties. UV–visible and emission spectra of P-1, P-2, P-3, and P-4 in different polar solvents are presented in Table 2 and Appendix A. As shown in Appendix A, P-1, P-2, P-3, and P-4 all have a broad absorption peak, and their maximum absorption wavelengths in NMP are 420, 420, 418, and 415 nm, respectively, which may correlate with the π–π* transition of the PCMB-D backbone. Based on an excitation wavelength of 418 nm, the maximum wavelengths of emission of P-1, P-2, P-3, and P-4 are 486, 486, 485, and 485 nm, respectively. The energy between the excitation and emission state of PCMB-D is slightly different, which is closely associated with the conjugated structure of PCMB-D. PCMB-Ds have good solubility in common organic solvents and large Stokes shifts. The Stokes shifts in different solvents are from 63 nm to 114 nm in Table 2. As we all know, a large Stokes shift (typically over 80 nm) is beneficial to reduce error and maintain the accuracy of detection [34]. The absorption wavelengths of P-1, P-2, P-3, and P-4 are firstly blue-shifting and then red-shifting with increasing solvent polarity, indicating that the charge transfers for four PCMB-Ds are almost the same. Particularly, PCMB-Ds in NMP exhibit excellent absorption and fluorescence properties (Appendix A). For example, the maximum absorption wavelengths of P-3 in EtOAc, THF, DMK, MT, DMF, and AN are 390.4, 412, 362, 363, 408, 362, and 418 nm, and the corresponding maximum absorbances are 0.181, 0.281, 0.149, 0.174, 0.225, 0.125, and 0.28, respectively, which may be due to NMP’s five-membered ring and strong polarity, which makes the π electrons more easily excited, and improves energy conduction between molecules. In addition, although polymer has limited solubility, PCMB-D in this experiment can be completely dissolved in NMP (Appendix A). Meanwhile, considering the good solubility of NMP for PVDF, NMP is selected as a good solvent for PCMB-D in this experiment, and P-3 appears as a bright yellow-green color in NMP as shown in Appendix A. At the same time, P-3 has an obvious fluorescence phenomenon whether it is powder or liquid, but Appendix A shows that the fluorescence intensity of P-3 in NMP decreases with the addition of water, which illustrates that P-3 has an aggregation-induced quenching effect. Additionally, the Mp values of M1-M5 are 113 °C, 104 °C, 94 °C, 91 °C, and 115 °C, respectively. As shown in Appendix A, PCMB-D only has a weight loss rate of 10% at about 400 °C and, in particular, P-3 can still maintain good condition at 41 °C, and retains 90% at 448 °C. Thereafter, the polymer backbone may degrade due to the high reactivity of C=C, and there is still a 50% carbon residue rate around 600 °C. In addition, P1, P-2, and P-4 also exhibit good thermal stability like P-3, which is closely related to their DP and narrow PDI, indicating that PCMB-Ds have the potential to be used as ECL materials. Overall, PCMB-Ds will be an appropriate electrode modification material in the field of ECL owing to their good luminous performance and stability.

### 3.2. Construction and Characterization of PCMB-D/ITO

After cleaning the ITO, PCMB-D/ITO was constructed by modifying PCMB-D and PVDF on ITO, and the ECL platform was formed with PCMB-D/ITO as the working electrode, platinum wire electrode as the auxiliary electrode, and Ag/Agcl as the reference electrode. The ECL performance of PCMB-D was similar, and P-3 was chosen as an example because of its best thermal stability, excellent photoluminescence, and ECL performance. As illustrated in Figure 1, the ECL signals could not be found without PCMB-D and TPrA, and showed a better signal when P-3-modified ITO was immersed in TPrA. This shows that P-3 and TPrA as co-reactants are indispensable for the ECL platform.

### 3.3. ECL Performance of P-3/ITO

The excellent thermal stability and optical properties of PCMB-D indicate that this type of material has the potential to be used as an ECL reagent. Unfortunately, when scanning cycles continued, the ECL strength of the PCMB-D decreased. The possible reason is the lack of strong force connection between the PCMB-D and ITO, causing the coating to peel off and making accurate detection difficult. In order to solve this problem, PVDF binder was introduced. PVDF and CPs were dissolved in NMP and modified on ITO. Using P-3 as an example, it was found that the cycle stability of the P-3/ITO can be significantly improved when the ratio of P-3 to PVDF is 7:3, shown in Figure 2a.

In order to further improve the ECL performance of P-3/ITO and obtain better ECL signals, the multi-parameter optimization of this ECL platform has been extensively studied. The effects of co-reactant, pH, number of modifier layers, and drying temperature on the solid-phase ECL sensor were mainly investigated. Figure 2b shows that P-3/ITO has no ECL signal in PBS and common co-reactants K_2_S_2_O_8_ and oxalic acid, but exhibits a clear ECL signal in the presence of TPrA. The optimal concentration of TPrA was also explored. P-3/ITO produced a superior ECL signal in 0.02 mol/L TPrA, as seen in Figure 2c.

In the measuring system, the pH environment of the buffer solution has a significant impact on the electron transfer rate of the electrode surface. As shown in Figure 2d, the ECL signal increases and then declines as the pH increases, and the highest value appears at about pH = 7. It was further found that pH = 7.4 was optimal for the formation of high-energy neutral amine radical reducing species.

In general, the ECL signal will be amplified with a higher concentration and bigger loading of P-3 on ITO. Nevertheless, as demonstrated in Figure 2e, more layers of P-3 modified on the ITO surface increase the resistance and decrease the ECL signal. A better signal is obtained by modifying a thin layer of P-3 on ITO with a thickness of 27 μm. In addition, as shown in Figure 2f, the ECL signal of P-3/ITO increased firstly and then decreased slightly with the increased drying temperature. As a result, the proper drying temperature was 120 °C.

The ECL performance of P-3 is evaluated under the above conditions. As shown in Figure 3a, P-3/ITO can emit an obvious ECL signal in 0.02 mol/L TPrA (0.1 mol/L PBS, pH = 7.4). Corresponding to ECL at an oxidation state of 1.0 V (Figure 3b), the excitation potential of P-3/ITO is 0.86 V (Figure 3c). The excitation potential of this type of polymer is relatively small, which shows that the P-3/ITO can be excited by a lower voltage. This can effectively save resources and reduce the damage to other components during the detection.

### 3.4. The Relationship between ECL Properties and the Structure of PCMB-D

To evaluate the effect of different alkyl side chains and other structural factors on the ECL properties, the ECL system of PCMB-D/ITO solid phase is constructed. As shown in 5b, the ECL response has good potential cycling stability for P-1, P-2, P-3, and P-4. In addition, Figure 4a,b show that the ECL signal intensity of P-3 is the best, which may be attributed to the alkyl side chains or DP.

To better understand the electronic properties and explore the influence of alkyl side chain length on ECL signal, density functional theory (DFT) calculations were performed [35]. Ground-state geometry optimization was carried out for P-1, P-2, P-3, and P-4 structures, which contain varying numbers of structural units (see structures in Figure 1b) at the ɷB97XD/6-311G(d,p) level [36,37]. As shown, the HOMO and LUMO of each structure basically overlap in space with each other, implying that there is no obvious intramolecular charge transfer phenomenon in these molecules [17], and thus proving that the PCMB-D/ITO is a co-reactive ECL system. In addition, the HOMO–LUMO gaps of the structural models of P-1, P-2, P-3, and P-4 show a very small variation, 3.827, 3.824, 3.824, and 3.823 eV, respectively (Appendix A). Therefore, the length of the side alkyl chain of PCMB-Ds shall impart limited influence on their ECL performance. Furthermore, in Figure 4 and Table 1, the signal of P-3 is the best, and that of P-4 is the weakest. The DP of P-3 (3.725 × 10^4^) is higher than that of P-4 (1.936 × 10^4^), which implies that higher DP of PCMB-D will help to improve the ECL performance. To further verify this conjecture, we compared the FL properties and ECL properties of three polyoctylcarbazole derivatives (P-3, P-3′, and P-3′’) with different DPs. As shown in Figure 5, the emission peaks of P-3, P-3′, and P-3′’ were 484 nm, 485 nm, and 487 nm at an excitation wavelength of 418 nm, respectively. A slight red shift was found, and this could be attributed to reduced DP. Moreover, the FL intensity and ECL signal of P-3 were the best, which indicated that the larger DP was beneficial to obtaining a better ECL signal, and the alkyl side chain had a small influence. In addition, the FL and ECL intensity of P-3′’ was the weakest, but it was still fluorescent material. This further verified that a higher DP can obtain a better FL and ECL signal. This supplies an idea for the future directional synthesis of CP ECL materials with better performance.

### 3.5. ECL Detection of Fe^3+^ with the P-3/ITO Sensor

From the above analysis results, P-3 has better thermal stability and ECL properties than other PCMB-D; in addition, the 9th position of P-3 is easily attacked, and the excited state of P-3 may undergo a redox reaction with strong oxidation of Fe^3+^, so a P-3/ITO solid-phase ECL sensing platform was constructed for subsequent sensing tests of Fe^3+^. In Figure 6a, the ECL intensity of P-3/ITO gradually declines after adding Fe^3+^ to the reaction solution, and the red-brown precipitation was found at the same time. The ECL signal of P-3/ITO was linearly fitted to the logarithm of concentration of Fe^3+^ (from 6 × 10^−8^ mol/L to 1 × 10^−5^ mol/L), and the linear equation was obtained from Figure 6b: (1)$${I=-\;2081\;-\;618\;\log\;c\;(\mathrm{Fe}}^{3+}{)\;(R}^{2}=0.954)$$
where ${\log\;c\;(\mathrm{Fe}}^{3+})\;$is the logarithm of concentration of Fe^3+^, and I is the ECL intensity of P-3/ITO to the concentration of Fe^3+^.

Furthermore, this limit of detection (LOD) was calculated according to the following formula:(2)LOD=3 σ / k
where σ is the deviation of response values, and k is the slope of the standard curve.

The calculated LOD is 2 × 10^−8^ mol/L, allowing for quantitative detection of Fe^3+^ from 6 × 10^−8^ mol/L to 1 × 10^−5^ mol/L. These data showed that the P-3/ITO platform is able to meet the detection requirements of Fe^3+^ in drinking water (5.36 μmol/L), in the environment (0.3 mg/L), and in serum (70~150 μg/L).

Compared with many methods reported for Fe^3+^ detection in many studies, our method has a wider detection range and lower limit, as shown in Table 3.

### 3.6. Stability and Selectivity of P-3/ITO Sensors

In order to explore the stability of P-3/ITO, this experiment recorded the ECL curve of P-3/ITO in 0.02 mol/L TPrA (0.1 mol/L PBS, pH = 7.4) under long-time scanning. The result is shown in Figure 7a that the ECL signal gradually increased until 140 s when the P-3/ITO was energized. Then, the ECL intensity reached the maximum value and maintained stability after 25 cycles of continuous scanning for 9 min. This indicates that the P-3/ITO ECL sensor has good stability.

The selectivity of the ECL sensor for the analyte is important. In this work, possible interfering substances in river water and serum were selected to explore the selectivity of P-3/ITO for Fe^3+^ detection, including NaCl, KCl, CuSO_4_, MgCl_2_, ZnCl_2_, CaCl_2_, Na_2_CO_3,_ and glucose with a concentration of 100 μmol/L. The results are shown in Figure 7b that the ECL intensity of P-3/ITO was obviously quenched in the presence of Fe^3+^ (10 μmol/L) compared to other substances, including Na^+^, K^+^, Cu^2+^, Ca^2+^, Mg^2+^, Zn^2+^, CO_3_^2+,^ and glucose. This indicated that the ECL sensor has good selectivity for Fe^3+^ detection.

With the increase in Fe^3+^ concentration, the ECL signal intensity of P-3/ITO gradually decreased, and the solution gradually became turbid until precipitation appeared. At the same time, the unique irritating odor of TPrA gradually disappeared. When the precipitate was removed, the ECL signal of the supernatant was still weak, indicating that the signal decay process was irreversible. As a result, the ECL decline was attributed to the consumption of the co-reactant TPrA or the luminescent material rather than the darkening of the electrode by the precipitate. Considering the stability of P-3 and the weak alkalinity of the reaction environment, a possible mechanism was proposed regarding the reduction in the ECL signal of P-3/ITO. Fe^3+^ and P-3 will compete for the co-reactant TPrA, but Fe^3+^ will preferentially transfer electrons to TPrA and form Fe^2+^, consuming the co-reactant. Next, there may be two pathways for further reducing the ECL signal: (1) Fe^3+^ will form a complex with P-3, which consumes the luminescent material and reduces the ECL signal. (2) In the presence of organic bases such as TPrA, Fe^3+^ may produce the precipitation of Fe (OH)_3_, which will further consume TPrA.

In order to further explore the experimental mechanism, the precipitates were purified with NMP, ethanol, and H_2_O, and analyzed with XRD and IR. It can be seen from Figure 8a that the precipitate has an amorphous structure, indicating that the precipitate is not Fe (OH)_3_. Furthermore, it can be seen from Figure 8b that the precipitate contains organic groups such as C=O and C-H, and alkyl stretching vibrations of C-H at 2929 cm^−1^ disappeared, which indicated that the second guess may be right, and implied that the precipitate was a complex formed by Fe^3+^ and P-3.

The generation of and change in ECL on this platform go through two stages (P-3 is represented as P).

Stage 1:

The excited state of P was formed via transferring electrons from TPrA to $P^{+\cdot }$.
(3)$$P\;-\;e^{-}\;\to {\;P}^{+\cdot }$$
(4)$$\mathrm{TPrA}\;-\;e^{-}\;\to {\;\mathrm{TPrA}}^{+\cdot }$$
(5)$${\mathrm{TPrA}}^{+\cdot }-\;H\;\to {\;\mathrm{TPrA}}^{\cdot }$$
(6)$$P^{+\cdot }+{\mathrm{TPrA}}^{\cdot }\;\to {\;\mathrm{TPrA}+P}^{*}$$
(7)$$P^{*}\;\to \;P+\mathrm{hv}$$

Stage 2:

In the presence of Fe^3+^, TPrA was preferentially consumed by Fe^3+^, and P can combine with Fe^3+^ to form a complex.
(8)$${\mathrm{TPrA}}^{\cdot }{+\mathrm{Fe}}^{3+}\;\to {\;\mathrm{TPrA}+\mathrm{Fe}}^{2+}$$

After forming a complex, the luminescent material was gradually consumed, therefore the ECL signal of the platform gradually decreased.

The reaction mechanism is shown in Figure 2.

## 4. Conclusions

In this work, four novel PCMB-Ds with different alkyl side chains have been designed and synthesized. All obtained PCMB-Ds display good solubility and thermal stability, and can emit a strong ECL signal in the presence of TPrA, which provided an ideal alternative polymer ECL reagent. In addition, we explored the relationship between the structure of PCMB-Ds and their ECL performance. Meanwhile, we found that the ECL signal of PCMB-D is enhanced with increasing DP, and the length of the alkyl side chain had little effect. It is proved that the structure of PCMB-D has a certain influence on its ECL performance, which will provide ideas for the preparation of polymer ECL materials in the future. Moreover, based on the preferred P-3, a solid-phase ECL sensor has been constructed, which presents high selectivity to Fe^3+^ and the LOD is 2 × 10^−8^ mol/L. Furthermore, the mechanism is explored that Fe^3+^ and P-3 competed for the TPrA during the detection process and causes the ECL signal change in P-3/ITO. To our knowledge, this is the first report of polycarbazole materials used in ECL sensors, which will open a new avenue for polycarbazole materials in the field of ECL analysis.

## Data Availability

The data used to support the findings of this study are available from the corresponding author upon request.

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
