# Peer review of "Poly(carbazole-co-1,4-dimethoxybenzene): Synthesis, Electrochemiluminescence Performance, and Application in Detection of Fe3+"

_polymers, 2022, doi:10.3390/polym14153045_

Round 1

Reviewer 1 Report

This manuscript describes the synthesis, characterization and Fe(III) ion sensing application of four poly(carbazole-co-1,4-dimethoxybenzene) copolymers. Besides, the authors performed DFT calculations to determine the energy levels of the polymers. The authors concluded that P-3 based solid-phase ECL sensor can efficiently detect Fe(III) ions. I support the publication of this work in Polymers after following corrections/changes.

 1.      Scheme 1 should be revised. Simply replacing different alkyl groups with “R” can save a lot of space and make the scheme clearer. Special care should be given to the bond length and angle.

2.      My major concern is regarding the analytical data. NMR Spectra given in the SI are not clean and show that the compound has a lot of impurities. For example, monomers M1, M2 & M4 and polymers P1-P4 have several additional unassigned peaks. Also, the multiplicities are not as expected. Therefore, I suggest authors recollect the NMR and include well-integrated spectra.  13C NMR data should also be included. All spectra should be in the same format and scale.

3.      In the IR spectrum, authors are suggested to assign relevant peaks (i.e. C=C, C-Har, C-Hal). Also, it would be better if overlaid monomer and polymer spectra are included.It is not clear what are the peaks ~3500 cm-1

4.      In UV-Vis spectra, draw the plot on a full 200-800 nm scale (as the λmax. below 400 given in Table 1 is not visible). Define all the abbreviations and check symbols in Table 1. 

5.      What is the excitation wavelength used in emission studies? Is there any aggregation effect? 

6.      Fig S15: What is the wavelength of UV light? Also, what was the solvent used as it seems that polymer has limited solubility! 

7.      I do not see any discussion on NMR, IR, photophysical and TGA (% loss and decomposition steps) in the manuscript. Authors are advised to include all in the revised MS. 

8.       I do not understand the use of DFT calculation as there is hardly any discussion on FMOs distribution and structural parameters.

9.      Careful revision in the English language is required.

Reviewer 2 Report

Dear Editor,

I have read the manuscript entitled: “Poly(carbazole-co-1,4-dimethoxybenzene): synthesis, electro-2 chemiluminescence performance and application in detection 3 of Fe3+

” and I would like to address following suggestions to the authors:

Scheme 1a should be enlarged.

Should the authors discuss the Stokes shifts data from Table 1 in the text?

Table 1: What is the difference between "1 The maximum absorption wavelength" and "2 The maximal absorption of PCMB-D", could the authors express it differently or give more details? This should be clarified in the text.

Figure 5a: In the legend of Fig. 5a the authors could add the value of the excitation wavelength used to record the emission spectra. Sample P-3" is non-fluorescent, could the authors explain? This should be clarified in the text.

How were the values for limit of detection (LOD) calculated?

The authors should check and rewrite the captions for Figs 6b and 7b.

I ask the authors to check spelling and others grammatical errors:

Line 22, Deficiency or excessive of Fe3+= Deficiency or excessive use of Fe3+; Line 88, PVDF was acted =PVDF acted; Line 105, facilitates to observe = facilitates observations; Line 138,  c2.4. = 2.4.; Line 149, 3.1. synthesis = 3.1. synthesis ; Line 155, the absorption = The absorption ; Line 209, value appeared = value appears; Line 271, strongly oxidizion = strongly oxidation ;

Round 2

Reviewer 1 Report

The authors have revised the manuscript and addressed almost all the queries of the reviewers. After going through the manuscript, I have the following suggestions in the manuscript.

1. Scheme 1a: Reaction condition (reagents, solvent, time, temperature, etc.) is missing for all the steps. Instead of writing CRH2R+1, authors are suggested to write “R” only (Where R = C2H5, C4H9, C8H17, and C16H33)

2. Authors are suggested to transfer Scheme 1(b) (i.e. HOMO/LUMO) to the section where it has been discussed. In the current version, it does not make any sense to combine synthetic scheme with the FMOs.

3. Some peaks in the NMR are still not assigned correctly. For e.g. I cannot see DMSO peak in the M1 NMR spectrum; for the same compound, there is no triplet at δ1.25 ppm and quartet at δ4.47 ppm in the spectrum. Authors are strongly advised to recheck the spectra and data for all the compounds.

4. Similarly in the IR spectrum, Csp2-H stretching is missing. Also, a strong peak around ~1500 cm-1 is not assigned. There are two C=C labels in the spectra which is causing confusion.

5. In DFT section, it is well established that the alkyl group at 9 positions of carbazole has a limited role in tuning the HOMO/LUMO level, so it is not very useful. If authors wish, they may include calculated absorption spectra, IR spectra etc.

6.      Language is not up to the mark.
